# Functional characterization of the 12p12.1 renal cancer-susceptibility locus implicates *BHLHE41*

Pierre Bigot[1,2], Leandro M. Colli[1], Mitchell J. Machiela[1], Lea Jessop[1], Timothy A. Myers[1], Julie Carrouget[2], Sarah Wagner[3], David Roberson[3], Caroline Eymerit[4], Daniel Henrion[5] & Stephen J. Chanock[1]

Genome-wide association studies have identified multiple renal cell carcinoma (RCC) susceptibility loci. Here, we use regional imputation and bioinformatics analysis of the 12p12.1 locus to identify the single-nucleotide polymorphism (SNP) rs7132434 as a potential functional variant. Luciferase assays demonstrate allele-specific regulatory activity and, together with data from electromobility shift assays, suggest allele-specific differences at rs7132434 for AP-1 transcription factor binding. In an analysis of The Cancer Genome Atlas data, SNPs highly correlated with rs7132434 show allele-specific differences in *BHLHE41* expression (trend $P$ value $= 6.3 \times 10^{-7}$). Cells overexpressing *BHLHE41* produce larger mouse xenograft tumours, while RNA-seq analysis reveals that constitutively increased *BHLHE41* induces expression of *IL-11*. We conclude that the RCC risk allele at 12p12.1 maps to rs7132434, a functional variant in an enhancer that upregulates *BHLHE41* expression which, in turn, induces *IL-11*, a member of the *IL-6* cytokine family.

[1] Division of Cancer Epidemiology and Genetics, National Cancer Institute, National Institutes of Health, 8717 Grovemont Circle, Bethesda, Maryland 20892, USA. [2] Department of Urology, Angers University Hospital, 4 rue Larrey, Angers 49100, France. [3] Cancer Genomics Research Laboratory, Frederick National Laboratory for Cancer Research, Leidos Biomedical Research Inc., 8717 Grovemont Circle, Bethesda, Maryland 20892, USA. [4] Department of Pathology, Angers University Hospital, 4 rue Larrey, Angers 49100, France. [5] CNRS UMR 6214, INSERM U1083, Université d'Angers, UFR de Médecine, rue haute de reculée, Angers 49045, France. Correspondence and requests for materials should be addressed to S.J.C. (email: chanocks@mail.nih.gov).

Renal cell carcinoma (RCC) represents 2–3% of cancer worldwide, with the highest incidence in Western countries[1]. Hypertension, obesity, smoking and a first-degree relative with RCC have been identified as risk factors for sporadic RCC[2]. Studies of hereditary kidney cancer syndromes have identified a series of rare, highly penetrant mutations in key genes, many of which have provided new insights into drivers of renal cancer, including, *VHL*, *MET*, *FCLN*, *TSC1*, *TSC2*, *FH* and *SDH*[3]. In particular, these studies have revealed the importance of alterations in metabolic pathways in RCC, which, in turn, have led to the development of specific, targeted therapies, increasing therapeutic options while improving prognosis overall[4,5].

Genome-wide association studies (GWAS) have emerged as an important tool for discovering regions in the genome associated with cancer susceptibility. Surveying hundreds of thousands of correlated common SNP markers, GWAS have identified hundreds of regions of the genome that are associated with disease risk, and, in particular, >500 independent regions associated with more than two dozen distinct cancers[6,7]. Further work is required to fine map and investigate the functional basis of the susceptibility locus, which usually reveals perturbations in redundant regulatory pathways[7,8]. So far, for RCC, six susceptibility loci have been identified: 2p21, 8q24.21, 11q13.3, 2q22.3, 12q24 and 12p12.1 (refs 9–12) but many more are expected with the completion of the current international RCC GWAS project. Of the six susceptibility loci, the biological underpinnings have been explained for only one locus. At the 11q13 locus, Schödel *et al.*[13] showed that a protective haplotype reduces HIF-1 binding to a promoter-distal enhancer of cyclin D1 (*CCND1*).

Herein, we report on the investigation of a RCC susceptibility locus on 12p12.1, marked by two reported correlated variants, rs718314 and rs1049380, ($r^2 = 0.57$ and D' = 0.77, CEU)[12], which has been confirmed in an independent study in Europeans[14]. Interestingly, rs718314 has been implicated in a large, international consortia GWAS of the waist-to-hip ratio phenotype[15–17]. It is notable that the same risk allele increases the waist-to-hip ratio and susceptibility to RCC; an observation that suggests a shared mechanism for RCC and obesity, an established risk factor for RCC[2].

## Results

**Fine mapping of the 12p12.1 locus.** Individual genotypes within a 2 Mb region flanking rs718314 (chr12:25344550-27344550, GRCh36) were imputed for three studies from Purdue *et al.*[11] (IARC, NCI and WTCCC). Analysis was restricted to imputed variants with high confidence (info score $\geq$ 0.50), yielding 44 SNPs that retained significant association with RCC risk (score test $P$ value $< 5 \times 10^{-5}$; Supplementary Table 1). The two previously published GWAS SNPs, rs718314 and rs1049380, were strongly associated with RCC risk in our analysis[12] (score test $P$ value $= 3.44 \times 10^{-6}$ and $P$ value $= 5.27 \times 10^{-6}$). We performed a conditional analysis in the 12p12.1 region based on rs718314, and did not find evidence for an additional independent signal. The majority of the correlated, untested RCC SNPs localized to a non-coding region, which also overlaps with the 3′-UTR of the inositol 1,4,5-triphosphate receptor type 2 (*ITPR2*), a member of the second messenger intracellular calcium release channels and nearest plausible candidate gene (Fig. 1).

**rs7132434 is located in a transcriptional enhancer.** Four of the 44 variants with the highest $P$ values were located in 3 areas that exhibited chromatin patterns consistent with enhancers based on data from the ENCODE Project[18], which showed enrichment for H3K4me1 and H3K27ac histone modifications, DNaseI hypersensitivity, and multiple transcription factor ChIP signals (Fig. 1). Area A contains rs7132434 and rs10842707, area B contains rs718314 and area C contains rs12814794. We screened these four most promising SNPs, and one additional SNP also located in area B, rs17383134, in luciferase assays in three different human renal cancer cell lines (786-0, UO-31 and SN12-C). We chose to include rs17383134 in our functional studies even though the $P$ value for association with RCC was below our $5 \times 10^{-5}$ cutoff (score test $P$ value $= 9 \times 10^{-4}$) because of its location in a putative regulatory region, and because it showed linkage disequilibrium with the GWAS tag SNP rs718314 ($r^2 = 0.26$. D' $= 1$). Only the fragments which contained rs7132434 demonstrated allele-specific enhancer activity when cloned in both the forward and reverse orientation. This enhancer activity was stronger for the A allele of rs7132434, which is highly correlated with the C risk allele of rs718314 ($r^2 = 1$, CEU; Fig. 2a and Supplementary Fig. 1).

Analyses of electrophoretic mobility shift assays (EMSAs) with each of the five SNPs revealed that only DNA fragments containing rs7132434 produced a specific shift with nuclear extracts from renal cancer cell lines (UO-31 and 786-O); furthermore, binding was stronger for the A allele (Fig. 2b–d), consistent with the results from the luciferase assays. These data indicate that rs7132434 is located in an enhancer region and displays allele-specific preference for the risk-associated A allele.

**rs7132434 is located in an AP-1-binding site.** To identify which transcription factors bind rs7132434, we first performed *in silico* analyses of motif sequences with Genomatrix software, and then, extracted ChIP-seq data from the ENCODE Project, performed super shift EMSAs and ChIP. An investigation of sequence motif data suggests rs7132434 is located in an AP-1 transcription factor-binding site (Fig. 3a). AP-1 is a heterodimeric transcription factor composed of bZIP domain containing proteins like Fos and Jun[19,20]. ENCODE Project ChIP-seq data from MCF10A-Er-Src breast cells show that c-Fos binds to this site. To determine if AP-1 binds rs7132434 in renal cells, we performed competitive binding EMSAs with consensus sequences of candidate transcription factor-binding sites and nuclear extracts from three renal cancer cell lines. Only the AP-1 consensus site competed with binding to the probe containing rs7132434 (Fig. 3b). To verify that AP-1 was binding rs7132434, we performed super shift EMSAs with antibodies to JunB, c-Jun and JunD. Antibodies for c-Jun and JunD both produced a strong super shift of the complex, whereas the antibody to JunB barely suggested a shift (Fig. 3c and Supplementary Fig. 2a,b). Because of the significant homology at the protein level between c-Jun and JunD, reactivity of anti-JunD to c-Jun or vice versa, could produce the result we observed with either antibody. We repeated the super shift EMSAs with antibodies to these two proteins that were raised against regions that were less well conserved (see the 'Methods' section), and both antibodies again produced a super shift (Supplementary Fig. 2b, lanes 2 and 4). We reviewed expression of *JUN* (which encodes c-Jun) and *JUND* in renal tissue samples from TCGA and found that *JUN* expression is higher than *JUND* in both normal and tumour samples[21] (Supplementary Fig. 3). Because of the relatively higher expression of *JUN* than JunD, we performed chromatin immunoprecipitation (ChIP) with antibodies against c-Jun to verify *in vivo* binding of AP-1 to rs7132434. In UO-31 renal cancer cells, there is a nearly 10-fold enrichment (Student's paired $t$-test $P$ value $= 0.026$) for c-Jun relative to IgG control (Fig. 3d, Supplementary Fig. 4). Taken together these data indicate

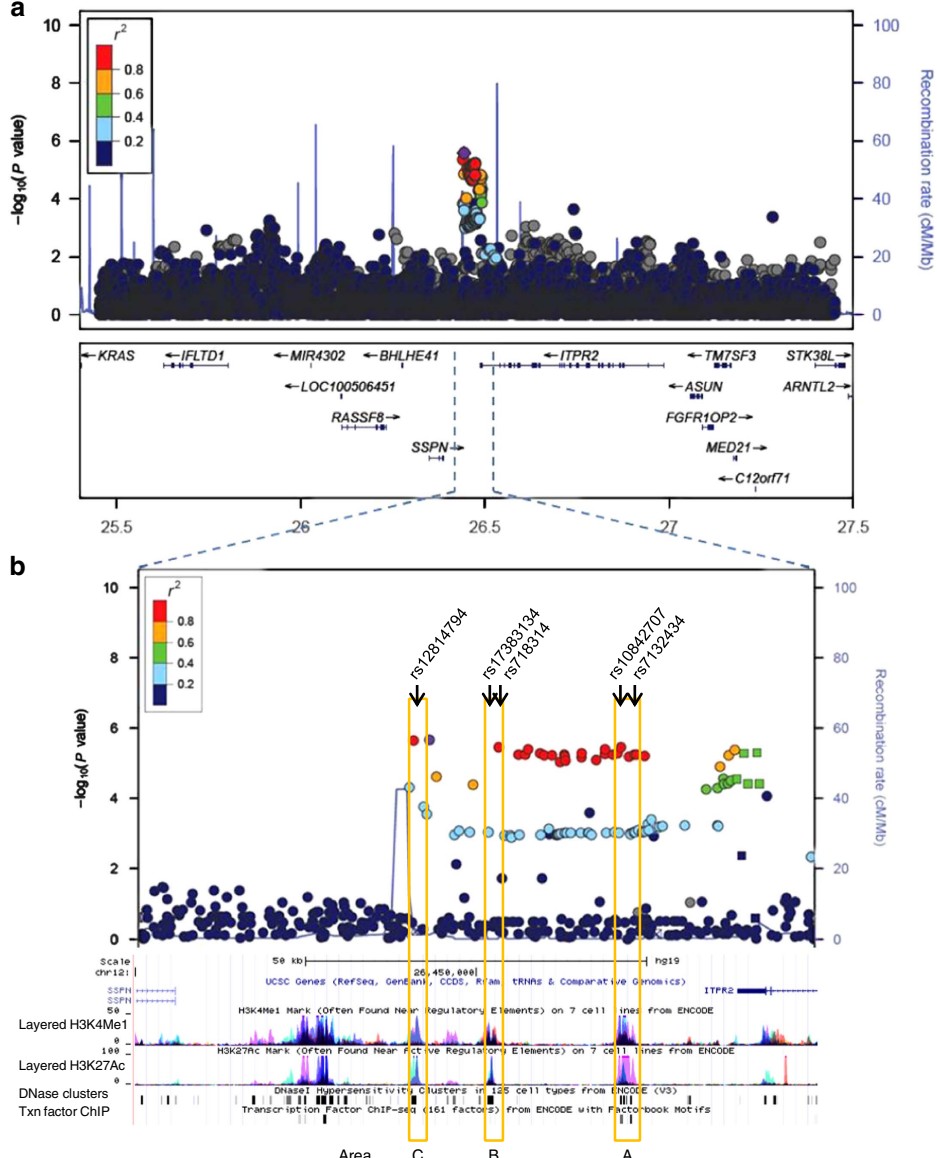

**Figure 1 | Association analysis and recombination hotspots of 12p12.1 RCC susceptibility locus. (a)** Combined meta-analysis $P$ values for association testing from 3 renal cancer scans within a 2-Mb region flanking rs718314 (chr12:25344550-27344550; GRCh36). **(b)** Enlarged view of the region of interest and peaks of H3K4Me1 and H3K27ac chromatin modification, DNase clusters and transcription factor-binding sites[18]. Area A contains rs7132434 and rs10842707, area B contains rs718314 and rs17383134; and area C contains rs12814794.

rs7132434 is located in an enhancer region in which the A allele (RCC risk allele) preferentially binds AP-1.

**eQTL analysis shows rs7132434 influences *BHLHE41* expression.** The original GWAS signal, rs718314, and our putative functional variant, rs7132434, were not genotyped in TCGA, but using LDlink[22], we identified rs10842708 as the best surrogate SNP ($r^2 = 1$ for rs7132434 and rs718314, CEU) genotyped in TCGA. The G allele of rs10842708 corresponds to the RCC risk allele and is strongly associated with increased expression of basic helix-loop-helix family, member E41 (*BHLHE41*, also known as *SHARP1*, *DEC2* and *BHLHB3*; trend test $P$ value $= 6.3 \times 10^{-7}$) in RCC, but not in normal kidney tissue (Supplementary Fig. 5a). Similarly, allele-specific expression differences were observed between rs10842708 and genes in close proximity on 12p12.1, namely, *SSPN* (trend test $P$ value $= 2.5 \times 10^{-4}$), *ITPR2* (trend test $P$ value $= 0.003$) and *RASSF8* (trend test $P$ value $= 0.009$; Table 1 and Supplementary Fig. 5b–d) in RCC, although the

associations were substantially less than for *BHLHE41*. The eQTL between rs10842708 and *BHLHE41* is specific to clear cell RCC (ccRCC) and is not observed in papillary RCC TCGA samples (Supplementary Fig. 6). In the TCGA samples, overexpression of *BHLHE41* in tumour is specific to RCC. We did not see higher *BHLHE41* expression in tumours from bladder, breast, colon, head/neck, liver, lung, pancreas or prostate (Supplementary Fig. 6e).

We did not observe an association between *BHLHE41* mRNA expression levels and adverse pathologic factors: Fuhrman grade (Wald test $P$ value $= 0.88$), metastasis (Wald test $P$ value $= 0.8$), node involvement (Wald test $P$ value $= 0.72$) or tumour stage (Wald test $P$ value $= 0.51$). In addition, we found no evidence that *BHLHE41* expression was associated with survival (log-rank $P$ value $= 0.923$; Supplementary Fig. 7) in ccRCC.

**BHLHE41 promotes tumour growth *in vivo*.** We knocked down the expression of *BHLHE41* by siRNA in a number of renal

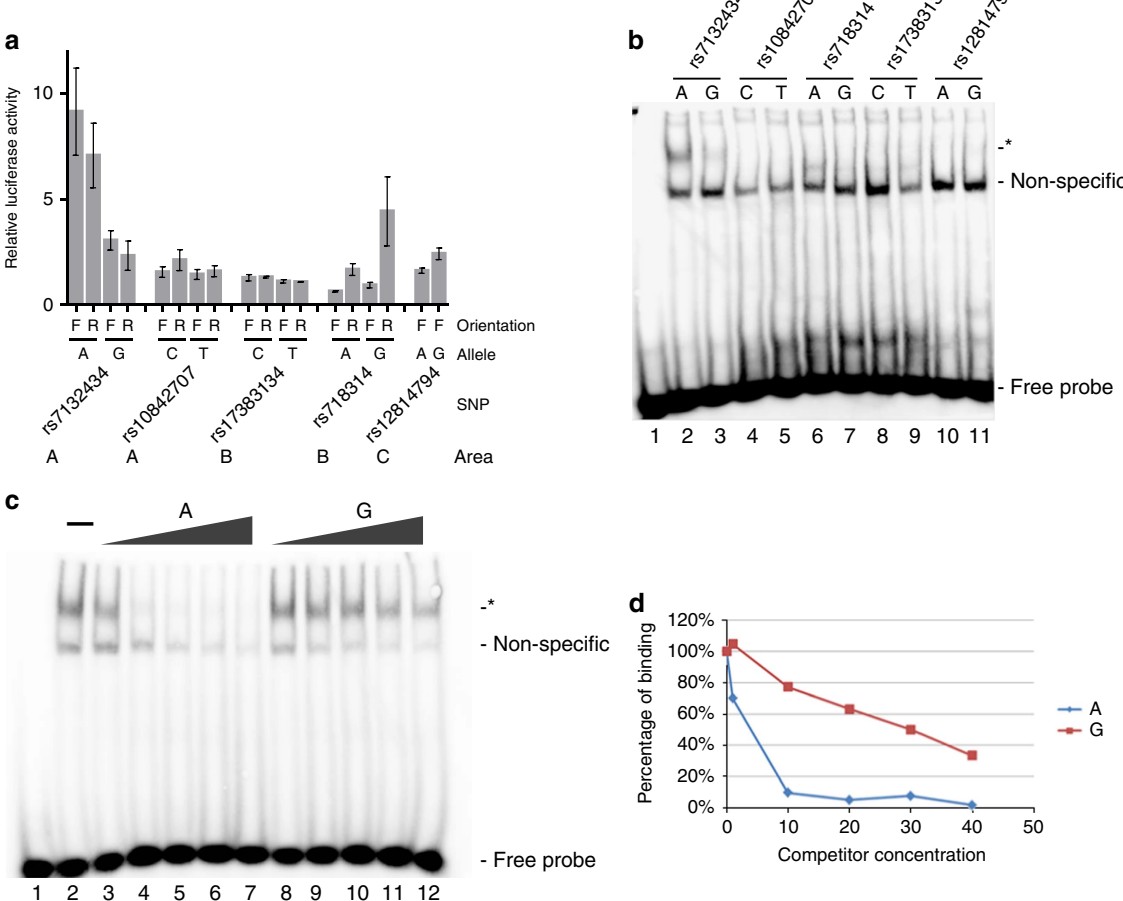

**Figure 2 | rs7132434 is in a functional enhancer.** (**a**) Luciferase assays for 5 promising SNPs associated with RCC with potential regulatory activity as per data from the ENCODE project tested in the 786-O renal cancer cell line. DNA fragments containing the indicated allele of each of the 5 SNPs were cloned in a forward (F) or reverse (R) orientation upstream of a minimal promoter that drives luciferase expression. A–C refers to the areas indicated in Fig. 1. The average relative luciferase activity of 3 independent experiments is plotted, independent experiments were performed in triplicate; error bars represent the standard deviation. (**b**) EMSA of the 5 SNPs with nuclear extracts of UO-31 renal cancer cell line. Lane 1 contains no extract. Each SNP was tested by EMSAs at least 3 times. Asterisk (*) denotes allele-specific differences in protein binding. (**c**) EMSA of probe containing the risk allele (A) of rs7132434 with nuclear extract from UO-31 cells. A 1 to 40-fold excess of unlabelled competitor with the indicated allele of rs7132434 was added to the reactions. Lane 1 contains no extract. Lane 2 contains extract, but no competitor. (**d**) Graph to the right shows percent bound.

cancer cell lines by transient transfection but did not observe a difference in growth rate or cell migration under either hypoxic or normoxic conditions (Supplementary Fig. 8a–c). In addition, neither transient, nor stable overexpression of *BHLHE41* in the renal cancer cell lines we tested, resulted in an observable phenotype (Supplementary Fig. 8d–f). Since the controlled environment of cells cultured in the lab can differ from those in a tumour, we investigated a xenograft model to further assess the role of *BHLHE41* in tumour formation and progression. Because TCGA data revealed that *BHLHE41* expression is higher in tumours, we performed xenografts with ACHN cells stably overexpressing *BHLHE41*. Five-week-old, female, nude, athymic BALB/c mice were injected in the right flank with $2 \times 10^6$ renal cancer ACHN cells that either stably overexpressed *BHLHE41* (ACHN-*BHLHE41*), or were stably transfected with empty control vector (ACHN-vector). ACHN-*BHLHE41* cells produced tumours that grew at a faster rate (ANOVA *P* value = 0.002, Fig. 4a) and, at the end of 6 weeks, had higher mass than tumours from ACHN-vector cells (281 versus 193 mg, *t*-test *P* value = 0.036; Fig. 4b and Supplementary Fig. 9). Less necrosis, as a per cent of surface area, was present in the tumours from ACHN-*BHLHE41* cells (*t*-test *P* value = 0.021; Fig. 4c,d). Sections of the tumours were immunostained to examine micro-vessel

formation by CD31 staining, and proliferative activity by KI-67 staining. Micro-vessel density was higher in tumours from ACHN-*BHLHE41* cells (*t*-test *P* value = 0.042, Fig. 4e), but the proliferative activity was lower (*t*-test *P* value = 0.014; Fig. 4f). Finally, immunohistochemical analysis of the xenograft tumours did not show a notable difference in the level of nuclear Hif-1 in the *BHLHE41* overexpressed tumours (71.4% versus 38.9%, *t*-test *P* value = 0.067).

**BHLHE41 induces expression of IL-11.** To investigate further how *BHLHE41* overexpression could promote tumour growth, we performed RNA-seq on ACHN-*BHLHE41* and ACHN-vector cells. There were 142 genes whose expression was altered by overexpression of *BHLHE1*; 107 of them were downregulated (Supplementary Table 2), consistent with reports that *BHLHE41* acts as a transcriptional repressor[23]. Pathway analysis, using DAVID[24,25] showed that 16% of the 142 genes were involved in the regulation of cell proliferation with 8.8% in positive regulation of cell proliferation.

We chose four genes for validation by quantitative real-time PCR: *BIRC3* and *EDN2*, both downregulated, and *SERPINE1* and *IL-11*, both upregulated by *BHLHE41* in ACHN. Consistent with

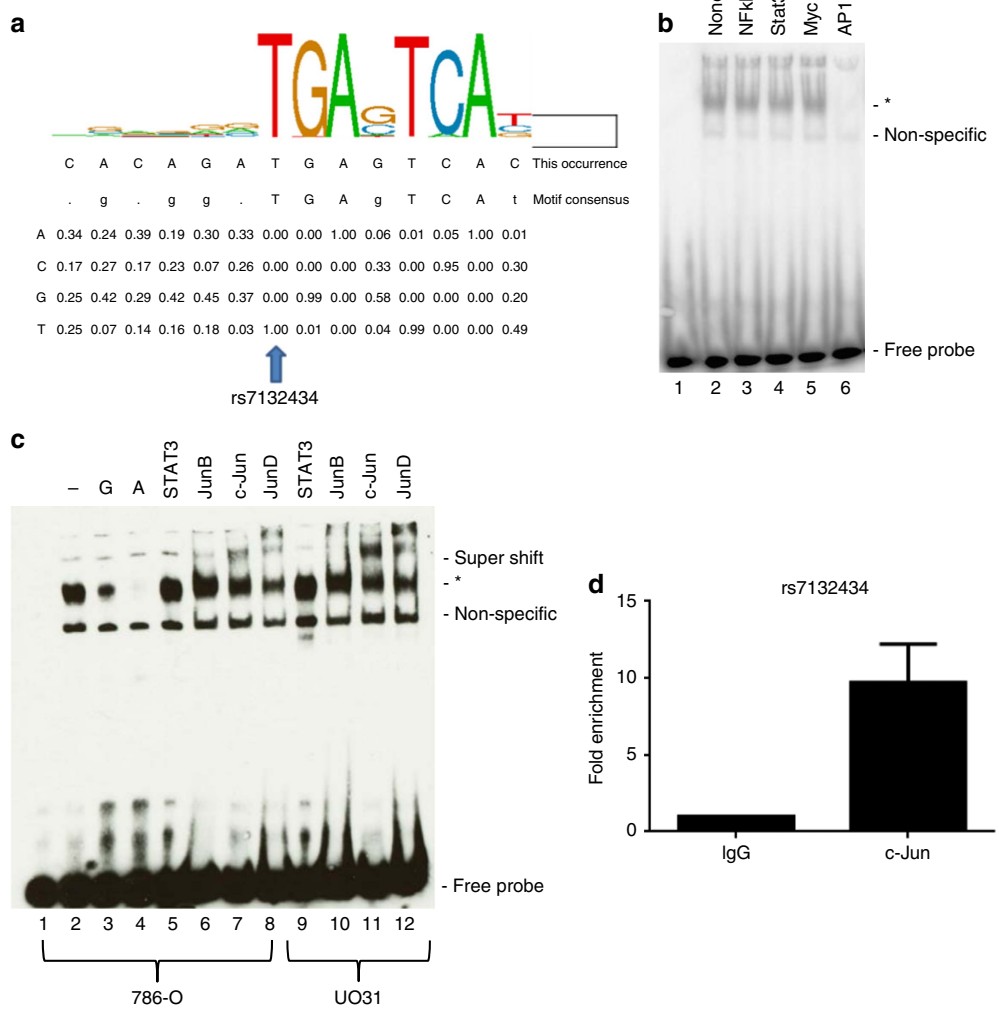

**Figure 3 | rs7132434 shows allele-specific AP-1-binding activity.** (**a**) AP-1 consensus binding site; rs7132434 changes a highly conserved base in the binding site. Lower panel is ChIP-seq results from ENCODE. (**b**) EMSA of rs7132434 with a 40-fold excess of unlabelled competitor with the indicated transcription factor-binding site added to the reactions. Lane 1 contains no extract. Lane 2 contains extract, but no competitor. Competition EMSAs were performed twice. Asterisk (*) denotes allele-specific differences in protein binding. (**c**) EMSA with nuclear extracts from 786-0 or UO-31 cells as indicated. Lanes 3 and 4 contain 40-fold excess of the unlabelled probe with the indicated allele of rs7132434. Lanes 5–12 contain the indicated antibodies. Super shifts were performed 3 times. (**d**) ChIP qPCR results using anti-c-Jun antibody (sc-1694) for rs7132434. Average fold enrichment relative to IgG for 3 independent experiments is plotted. Error bar represents the standard deviation. $P = 0.026$, Student's paired $t$-test, with two-tailed distribution. qPCR assays were performed in triplicate.

**Table 1 | Relationship between nominally significant SNP alleles and nearby gene expression.**

| SNP | R² with rs718314 (CEU) | MAF (CEU) | Association with RCC (P value) | eQTL association between genotype and gene expression (P value) | | | | | | | |
|---|---|---|---|---|---|---|---|---|---|---|---|
| | | | | ITPR2 | | RASSF8 | | SSPN | | BHLHE41 | |
| | | | | Normal | Tumour | Normal | Tumour | Normal | Tumour | Normal | Tumour |
| rs10842708 | 1.00 | 0.23 | 6.30E − 06 | 0.08 | 3.30E − 03 | 0.62 | 9.21E − 03 | 0.50 | 2.47E − 04 | 0.48 | 6.35E − 07 |
| rs12814794 | 0.97 | 0.22 | 2.31E − 06 | 0.24 | 0.03 | 0.73 | 0.01 | 0.51 | 8.06E − 04 | 0.63 | 2.65E − 07 |
| rs11048447 | 0.62 | 0.31 | 2.41E − 05 | 0.96 | 0.05 | 0.86 | 0.02 | 0.34 | 0.01 | 0.32 | 0.00034 |
| rs1049376 | 0.57 | 0.23 | 5.00E − 06 | 0.18 | 0.04 | 0.83 | 0.10 | 0.84 | 4.76E − 03 | 0.45 | 7.35E − 04 |
| rs9442 | 0.44 | 0.33 | 2.86E − 05 | 0.96 | 9.70E − 03 | 0.81 | 0.24 | 0.91 | 0.05 | 0.64 | 3.60E − 03 |
| rs2570 | 0.39 | 0.33 | 3.88E − 05 | 0.94 | 0.02 | 0.79 | 0.24 | 0.84 | 0.21 | 0.52 | 0.03 |
| rs10842702 | 0.28 | 0.37 | 4.83E − 05 | 0.85 | 6.59E − 03 | 0.73 | 0.07 | 0.79 | 0.15 | 0.67 | 9.47E − 04 |

MAF, minor allele frequency; RCC, renal cell carcinoma; SNP, single-nucleotide polymorphism; CEU, Utah residents (CEPH) with Northern and Western ancestry; eQTL, expression quantitative trait loci. Outliers removed (outliers defined as 1.5 times the interquartile range). All SNPs investigated have trend test ($P < 5 \times 10^{-4}$) with renal cancer risk.

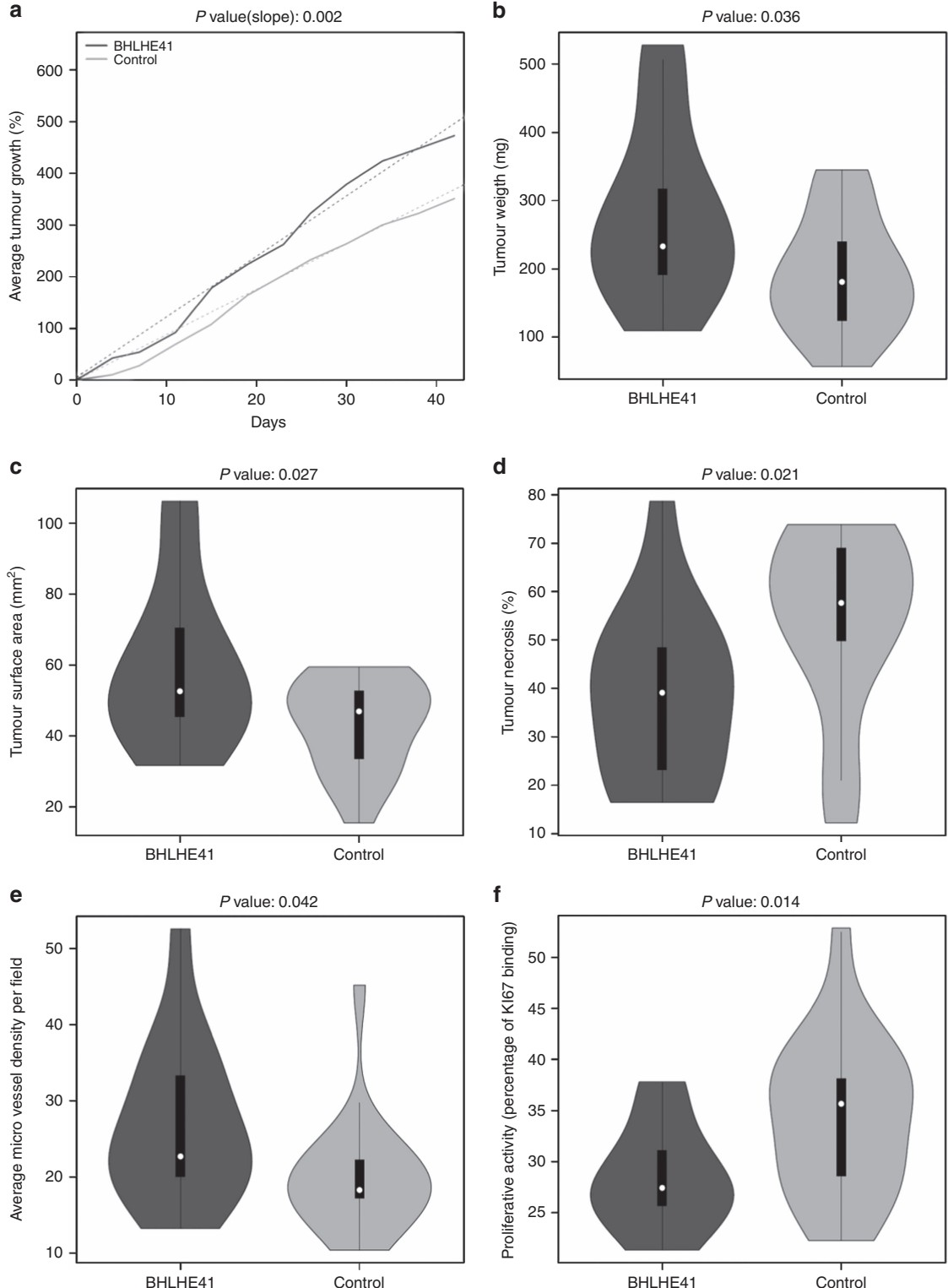

**Figure 4 | Cells overexpressing *BHLHE41* produce larger mouse xenografts.** Dark grey, tumours from 17 mice injected with ACHN-*BHLHE41* cells; light grey, from 20 mice injected with ACHN-vector. (**a**) Average tumour growth rate. Solid lines represent the group means; dashed lines are the best fit lines from the linear mixed model. For the violin plots in **b**–**f**, the white dot represents the median, thick black line is the interquartile range and the whiskers represent the data range. *P* values from a *t*-test are indicated. (**b**) Tumour weight. (**c**) Tumour surface area. (**d**) Tumour necrosis as a per cent of tumour surface. (**e**) Average micro-vessel density, as determined from CD31 staining. (**f**) Proliferative activity, as determined from KI-67 staining.

the RNA-seq data, increased *BHLHE41* expression in ACHN cells leads to an increase in the expression of *SERPINE1* and *IL-11*, while *EDN2* and *BIRC3* expression is decreased (Fig. 5). Similar effects were seen in a distinct renal cancer cell line, 786-0,

however, in HK2 cells (normal kidney) *SERPINE1* and *IL-11* were downregulated along with *EDN2*, while *BIRC3* was upregulated. We extended our analysis to a breast cancer cell line, MDA-231, and observed that *BHLHE41* induced *IL-11* up-regulation, and

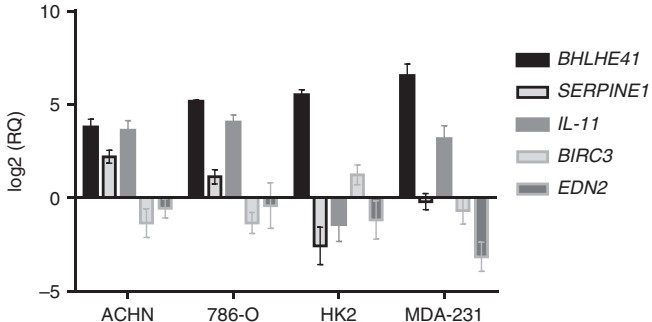

**Figure 5 | qRT-PCR validation of RNA-seq results.** Expression level of indicated genes was measured in renal cancer cell lines stably overexpressing *BHLHE41*. The log base 2 of the average relative quantity of the indicated genes from 3 independent experiments is plotted. Error bars represent the standard deviation. Each of 3 experiments was performed in triplicate.

*BIRC3* and *EDN2* down-regulation. There was little effect on *SERPINE1* (Fig. 5). Together with the above findings, we observe that *BHLHE41* contributes to the growth of RCC in a xenograft model, perhaps through induction of *IL-11*.

## Discussion

In this study, we investigated the biological underpinnings of the common susceptibility haplotype for RCC on chromosome 12p12.1. By testing potential functional SNPs in high LD at this RCC-associated locus, we identified rs7132434 as a functional SNP located in a transcriptional enhancer that binds AP-1. When the 12p12.1 renal susceptibility locus was discovered, because of its location in the 3′-UTR of *ITPR2,* it was hypothesized that *ITPR2* was the functional target of the GWAS signal[12]. *ITPR2* belongs to the family of inositol 1, 4.5-triphosphate receptors (IP3R) which are intracellular $Ca^{2+}$-release channels. These receptors function in cell division, cell proliferation, apoptosis, fertilization, development, memory and learning[26]. However, a recent eQTL analysis showed a correlation between rs718314 and *BHLHE41*, not *ITPR2* (ref. 27), and our results show that this locus has the strongest association with expression of *BHLHE41*.

*BHLHE41* (*SHARP1, DEC2, BHLHB3*) has been reported to be involved in the control of circadian rhythm, apoptosis and cell differentiation[23], and has also been shown to suppress breast cancer metastasis[28]. In kidney cancer patients evidence suggests that deregulation of genes involved in the circadian clock circuitry could influence disease progression and outcome[29]. Moreover, interaction between the endogenous circadian timing system and adipose function has recently been demonstrated[30], and further work suggests that inhibition of adipogenesis by *BHLHE41* through the regulation of C/EBP activity[31] provides a basis for the link between *BHLHE41* and obesity, a known RCC risk factor.

Transcriptional enhancer elements can be located distant from the target gene(s), and are brought into close proximity through DNA looping. This three-dimensional structure of the chromatin has been probed by chromosome-conformation-capture-related assays like Hi-C[32,33]. We examined the Hi-C data generated by Rao *et al.*[34] for evidence of physical interactions between rs7132434 and *BHLHE41*. We found that the SNP lies in a contact domain that contains *BHLHE41* in all 9 cell lines examined (Supplementary Fig. 10). This provides further evidence that rs7132434 is located in a functional enhancer of this gene.

Evidence from the cancer microarray database, ONCO-MINE[35,36], revealed microarray gene-expression studies

showing higher expression of *BHLHE41* in RCC tumours compared with adjacent normal kidney tissue, in the range of 7–29-fold more expression. In an analysis of TCGA RCC tumours, we observed marked overexpression of *BHLHE41*, but not in breast, lung, colon or other cancers (Supplementary Fig. 6e). Despite the high level of *BHLHE41* in RCC, we did not find any association between *BHLHE41* expression and classical adverse pathologic factors and reduced survival (Supplementary Fig. 7). This could be explained by small numbers and heterogeneity of ascertainment of cases, as well as the possibility that the 12p12.1 susceptibility locus contributes to RCC susceptibility, but not as robustly for RCC disease progression.

Overexpression or knockdown of *BHLHE41* in renal cancer cell lines had no discernible phenotype with respect to cell proliferation, or migration, under either normoxic or hypoxic conditions (Supplementary Fig. 8). However, when ACHN cells stably overexpressing *BHLHE41* were injected into the flanks of nude mice, tumours grew faster and larger than those from control ACHN cells (Fig. 4) demonstrating that *BHLHE41* may in fact have a role in promoting tumour formation and/or progression. Further studies will be needed to differentiate this latter issue.

Previously, Montagner *et al.*[28] showed that triple-negative breast cancer (TNBC) patients have better metastasis-free survival if their tumours express higher levels of *BHLHE41*. They went on to show that *BHLHE41* inhibits the invasive and metastatic phenotype of TNBC by promoting the proteasomal degradation of HIF, independently of VHL[28]. *BHLHE41* has also been shown to be important in the adaptation to hypobaric hypoxia. Huerta-Sanchez *et al.*[37] demonstrated that *BHLHE41* genetic variants are selected during hypoxic adaptation. In addition, the promoter region of *BHLHE41* contains a hypoxia response element that is bound and transcriptionally regulated by HIF generating an apparent negative-feedback loop[38]. The VHL/HIF pathway is known to have a central role in ccRCC[39], and Montagner *et al.*[28] showed that *BHLHE41* does act on HIF in some renal cancer cell lines. However, our data suggests that *BHLHE41* may also act in a manner independent of the regulation of HIF in RCC. First, the expression of *BHLHE41* is increased in ccRCC tumours, not decreased as is the case for TNBC. Second, we examined Hif-1α protein levels by western blot in a number of renal cancer cell lines (ACHN, UO-31 and HK2) and did not detect any change in the amount of Hif-1α when *BHLHE41* was overexpressed or knocked down by siRNA (Supplementary Fig. 11–13). Finally, immunohistochemical analysis of the xenograft tumours did not show an appreciable difference in nuclear Hif-1α protein levels when *BHLHE41* was overexpressed (71.4% versus 38.9%, *P* value = 0.067).

To better understand the function of *BHLHE41* in RCC we performed RNA-seq on the ACHN-*BHLHE41* extracted cells. *IL-11* is one of the genes exhibiting the greatest *BHLHE41* induced differences in expression. *IL-11* is a member of the *IL-6* cytokine family that signals through gp130 and IL-specific receptors. *IL-6* signalling has been shown to be an important pathway in RCC, by activating STAT3 which promotes tumorigenesis, inhibits apoptosis, and enhances proliferation, angiogenesis, invasiveness and immune evasion[40]. The direct role of *IL-11* in RCC has not been well studied, but it has been shown to stimulate tumour cell proliferation in colorectal adenocarcinoma[41] and overexpression is correlated with poor clinical outcome in gastric cancer[42]. Onnis *et al*[43] showed that *IL-11* activates an oncogenic signalling pathway, through STAT1 and 3, that increased anchorage-independent growth in prostate cancer PC3, colon cancer HCT116, and renal cancer RCC4 cells. In a xenograft model, knockdown of *IL-11* by shRNA in PC3 cells

produced smaller tumours than control[43]. Finally, increased *IL-11* level, measured by immunohistochemistry, was associated with an increased chance of ccRCC recurrence and decreased survival[44].

Our data suggest that rs7132434 is a functional variant residing on the RCC susceptibility haplotype identified by GWAS of RCC on 12p12.1. This variant lies in a transcriptional enhancer that binds AP-1 (ref. 20) with allele-preference for the RCC risk allele, A. We show that the enhancer acts on several genes in the region, with the strongest effect on *BHLHE41*, whose effect is complex and promotes renal cancer formation and progression. In a xenograft model, tumours overexpressing *BHLHE41* have higher growth rate and greater mass. One plausible mechanism for the role of *BHLHE41* in RCC may be through induction of IL-11, which activates STAT3[45] to promote cell cycle progression and inhibit apoptosis, although further experiments are needed.

## Methods

**Fine mapping of the 12p12.1 locus.** We performed an imputation analysis with three independent European samples from previously published GWAS of RCC[11]. The first scan included samples from the International Agency for Research on Cancer (IARC; 1,936 RCC cases and 3,742 controls), the U.S. National Cancer Institute (NCI; 1,311 RCC cases and 3,424 controls) and the Wellcome Trust Case Control Consortium (WTCCC; 350 RCC cases and 1,361 controls)[11]. Informed consent was obtained from study participants as well as approval from each study site's Institutional Review Board (see ref. 11 for further information).

To infer untested genotypes, we used IMPUTE2 (ref. 46), to impute SNPs ± 1 Mb of rs718314 based on the 1000 Genomes Project[47] and Division of Cancer Epidemiology and Genetics reference panel[48]. On the basis of per marker quality measures generated from IMPUTE2, markers with a maximum information score < 0.5 and a minor allele frequency < 0.01 were filtered. The genotyped and imputed SNP genotypes for all markers were tested for association with renal cancer risk using a score test assuming an additive genetic model in SNPTEST (https://mathgen.stats.ox.ac.uk/genetics_software/snptest/snptest.html). We performed principal component analysis for each population using the GLU struct.pca module. The final analysis was adjusted for gender and significant eigenvectors (three for IARC population, one for WTCCC and none for the NCI Population). Q–Q plots of the observed verses expected *P* value distribution were used to assess possible residual population structure. Effect estimates and standard errors were then combined across samples with a fixed-effects meta-analysis using METAL software[49]. A conditional analysis was performed based on the most significant initial SNP, rs718314.

**Cell culture.** All cell lines were grown in media supplemented with 100 U ml$^{-1}$ penicillin, 100 µg ml$^{-1}$ streptomycin and 10% FBS and maintained in a 37 °C incubator with 5% CO$_2$. HK2 (ATCC CRL-2190), SN12-C (NCI-60), UO-31 (NCI-60), ACHN (ATCC CRL-1611) and MDA-MB-231 (ATCC HTB-26) were grown in DMEM. 786-O (ATCC CRL-1932) was grown in RPMI. For hypoxia, dishes were placed in an airtight chamber and flushed with 1% O$_2$, 5% CO$_2$ daily and placed in a 37 °C incubator[50,51]. Cells were authenticated using the AmpFLSTR Identifiler PCR Amplification Kit (ThermoFisher). All cultures tested negative for mycoplasma contamination using the MycoAlert Mycoplasma Detection Kit (Lonza).

**Luciferase assays.** *BHLHE41* cDNA (NM_030762.1) from Origene. *BHLHE41*-myc cDNA was synthesized by NeoBioLabs and cloned into the pcDNA3.1/Hygro Mammalian Expression Vector (Thermo). DNA fragments used in the luciferase assays were PCR amplified using primers listed in Supplementary Table 3, and cloned into pGL4.23 (Promega). All cloned sequences were verified by Sanger sequencing.

Transfections were performed using Lipofectamine 2000 (Life Technologies) according to the manufacturer's instructions. After 24 h, lysates were collected and luciferase activity measured using the Dual-Luciferase Reporter Assay System (Promega). Experiments were carried out in triplicate.

**siRNA knockdown.** ON-TARGETplus Human *BHLHE41* siRNA–SMARTpool (L-010043-00) and ON-TARGETplus Non-targeting Control Pool siRNA (D-001810-10) were obtained from Thermo Scientific. Transfections were out carried using Lipofectamine RNAiMAX (Life Technologies) according to the manufacturer's instructions. Cells were collected after 24, 48 and 72 h. Cell proliferation assays were performed using the WST-1 Cell Proliferation Reagent (Roche), or FluoReporter Blue Fluometric dsDNA Quantitation Kit (Molecular Probes) according to the manufacturer's instructions.

**Electrophoretic mobility shift assays.** Nuclear extracts were prepared using the Nuclear Extract Kit (Active Motif) following manufacturer's instructions.

Complementary 3′-biotinalayted oligos (sequences in Supplementary Table 1; Life Technologies) were annealed to create substrates for EMSA using the LightShift Chemiluminescent EMSA Kit (Pierce). Reactions contained 170 ng of biotinalayted probe, 3 µg nuclear extract, 3 µg BSA, 10 µg sheared salmon sperm DNA (Sigma), 10 mM Tris, pH 7.5, 1 mM KCl and 2% glycerol. Competitive assays were performed by adding a 1–40-fold excess of non-biotinylated probes. Consensus binding sequences for specific transcription factors from Santa Cruz Biotechnology; sc-2505, NFkB; sc-2571, Stat3; sc-2509, Myc-Max; sc-2501, AP-1. The following antibodies (2 µl) were used for super shift experiments: ab31417 and ab31419, c-Jun; ab28837 and ab134067, JunD; ab31421, JunB; sc-7202, c-Fos; sc-150, C/EBP β; sc-372, NFkBp65; sc-66931, IgG; sc-482, Stat3. The abcam antibodies ab31417 and ab134067 were raised against peptides from c-Jun or JunD, respectively, that are less well conserved between the two proteins.

**cDNA isolation and gene-expression assays.** SuperScript III First-Strand Synthesis System (Life Technologies) was used to synthesize cDNA from 1 µg total RNA with random hexamers. Gene expression was measured with TaqMan Gene-Expression Assays (Life Technologies). For *BHLHE41*, assay Hs01036450_g1 was used. Four housekeeping genes, β2-microglobulin (*B2M*, Hs99999907_m1), glyceraldehyde-3-phosphate dehydrogenase (*GAPDH*, Hs99999905_m1), peptidyl-prolyl isomerase A (*PPIA*, Hs99999904_m1) and β-actin (*ACTB*, Hs99999903_m1), were used as endogenous controls.

**RNA-seq.** Total RNA was DNase-treated using Ambion's TURBO DNA-free, then purified using RNAClean XP (Agencourt/Beckman Coulter). Ribosomal RNA was removed using Epicentre's Ribo-Zero Gold (Human/mouse/rat), followed by RNAClean XP purification. Barcoded sequencing library preparation was performed using Kapa Stranded RNA-Seq library preparation kit (Kapa Biosystems). Libraries were quantitated for sequencing on the Agilent Bioanalyzer with the high sensitivity DNA kit. Sequencing was performed on the NextSeq 500 (2 × 151-bp paired end, v2 chemistry; 8 samples were run across two sequencing runs).

NextSeq 500 BCL to fastq conversion and de-multiplexing was performed using Illumina's BCL2fastq (v2.15), followed by lane and run merging of each sample on the cloud-based Seven Bridges Genomics platform. Sequencing quality was assessed using FastQC. Contaminating adaptor sequences were removed using FastqMcf (v1.4). Paired reads were aligned to hg19 using STAR aligner with UCSC annotations (May 2014 release), using default parameters and an overhang length of 149 bp. PicardMarkDuplicates removal PCR duplicate reads from aligned BAMs. Batch QC of all samples was performed using RNA-SeQC (Broad). Quantitative gene-expression analysis was performed using the Cufflinks package, and differential expression analysis was performed using CuffQuant and CuffDiff.

**Immunoblotting.** Antibodies were: anti-human HIF-1α (clone 54; BD Transduction Laboratories, San Jose, CA; 1:1,000), anti-HIF-2α (AF2886, R&D Systems, Minneapolis, MN; 1:1,000), anti-β-actin (NB600-505, Novus Biologicals, Littleton, CO; 1:5,000), anti-MYC (ab9106, AbCam, Cambridge, MA; 1:5,000) anti-BHLHE41 (a generous gift from Montager and Piccolo[28]; 1:100), goat anti-mouse HRP (NB7511, Novus Biologicals, Littleton, CO; 1:5,000), goat anti-rabbit HRP (NB7160, Novus Biologicals; 1:5,000) and donkey anti-goat HRP (sc-2020, Santa Cruz Biothenology, Dallas, TX; 1:1,000).

Whole-cell extracts prepared in RIPA buffer with protease inhibitors (cOmplete Protease Inhibitor, Roche) were electrophoresed on 4–12% Bis Tris Plus Bolt gels in MES buffer (Invitrogen). Proteins were transferred to nitrocellulose using an iBlot2 (Invitrogen). Blots were blocked with 5% non-fat dry milk in Tris-buffered saline with 0.1% Tween-20 (TBST). Primary and secondary antibodies were diluted in 5% milk in TBST, and all washes were performed with TBST. Blots were rinsed briefly with PBS before the addition of ECL Prime Western Blotting Detection Reagent (Amersham).

**Chromatin immunoprecipitation.** ChIP was performed using the Active Motif ChIP-IT High Sensitivity Kit. Antibodies were 4–10 µg of anti-c-Jun (ab31419, AbCam), 10 µg of c-Jun (sc-1694; Santa Cruz Biotechnology) and equivalent amounts of normal rabbit IgG control (12–370, EMD Millipore). Fold enrichment of the rs7132434 region was quantified by SYBR green real-time PCR (primers listed in Supplementary Table 3). All real-time PCR amplifications were performed in triplicate.

**Analysis of data from TCGA database.** Expression, genotyping and clinical data for the TCGA samples were downloaded from The Cancer Genome Atlas database (http://cancergenome.nih.gov/, accessed on 9 October 2013). RNA sequencing data from 481 tumours and 71 normal tissues were examined to assess any associations between the expression of 12p genes and the RCC-associated variants in the 12p12.1 region. We quantified expression as normalized read counts and removed outlier samples with expression values exceeding 1.5 times the inter-quartile range. Linear trend tests were used to test for allele-specific increases in gene expression. We examined RNA sequencing data from eight other TCGA tissues (bladder, breast, colon, head/neck, liver, lung, pancreas and prostate) to

assess *BHLHE41* expression in normal versus tumour tissues. Independent-sample *t*-tests were used for comparisons of expression values.

We used a multivariate regression model to search for an association between *BHLHE41* expression and Fuhrman grade, tumour stage, node involvement, age and the presence of metastasis at diagnosis. The outcome was determined from date of surgery to date of last follow-up. Cancer-specific survival (CSS) was estimated using the Kaplan–Meier method and a comparison was performed by log-rank test using the median value of *BHLHE41* expression as cutoff and the different genotypes of rs7132434. R version 2.15.3 software was used for all statistical analyses.

**Murine xenograft model.** Animal experiments for this study were approved by the institutional animal care and use ethics committee of Angers, France. Female nude athymic BALB/c mice, 5 weeks old (Janvier Laboratory, France), were injected subcutaneously in the right flank with $2 \times 10^6$ cells in 100 μl RPMI-1640 and 100 μl of Matrigel (Corning, Life science, Amsterdam, Pays-Bas). Twenty-one mice received ACHN-vector cells, and 21 ACHN-*BHLHE41* cells. Mice were selected at random. One mouse injected with ACHN-vector and 4 with ACHN-*BHLHE41* cells died 1 day after injection.

Tumour diameters were measured twice per week with electronic calipers, and tumour volumes were calculated using the formula: $((\text{width})^2 \times \text{length})/2$ (mm$^3$). The investigator performing this measurement was not blinded. The weight of the mice was measured twice per week. The mice were killed 6 weeks after the injection and the tumours were resected and subjected to immunohistochemical analysis. Two mice injected with ACHN-vector and three with ACHN-*BHLHE41* cells failed to produce tumours, were considered a failure of the xenograft and were excluded from the analysis.

**Histology and immunohistochemical.** Tumours were fixed in 10% formaldehyde solution and embedded in paraffin. Immunohistochemical staining was performed on 4-μm tumour sections. We used the automatic Bond III system and Bond Polymer Refine Detection (DS 9800; Leica Biosystems). Endogenous peroxidase activity was blocked by incubation with peroxide block solution. The primary antibodies used were anti-Hif-1α (ab8366, AbCam), anti-Ki-67 (clone MIB-1, Dako) and anti-PECAM1 (CD31; Santa Cruz biotechnology).

Micro-vessel density was evaluated with anti-PECAM1 using the method of Weidner *et al.*[52]. Micro-vessel density score was calculated as the mean of five areas. Proliferative activity was evaluated with anti-Ki-67. The percentage of positive tumour cells was calculated using four fields. All specimens were examined blindly by two pathologists.

**Statistical analysis.** Statistical analyses and plotting of xenograft results was performed in R 3.1.2 statistical software. Linear mixed models were fit to the tumour growth data with random effects for each mouse and fixed- effect terms were included for *BHLHE41* group and *BHLHE41* by day interaction. To test for a statistically significant effect of *BHLHE41* on growth, an ANOVA was used to compare the full model with a model with no *BHLHE41* covariates. All other statistical tests were two-sample *t*-tests, unless otherwise stated. Statistical tests were two-sided.

**Data availability.** The RNA-seq data have been deposited in the NCBI bioproject database under accession code PRJNA309249. Hi-C data from Rao *et al.*[34] can be found in the Gene Expression Omnibus (GEO) under accession number GSE63525. The RCC GWAS data have been deposited in dbGaP under the accession code phs000351.v1.p1. The TCGA data referenced in this study are available at http://cancergenome.nih.gov/. The authors declare that all other data are contained within the paper and the Supplementary files or available on request.

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

## Acknowledgements

This work is supported by the NIH intramural research program. P.B. was partially funded by the Association Française d'Urologie. We thank S. Piccolo and M. Montagner for their generous gift of *BHLHE41* rabbit polyclonal antibody.

## Author contributions

P.B. and S.J.C. conceived the project. P.B. and M.J.M. defined regions of interest and performed eQTL analysis. P.B. and L.M.C. designed and performed experiments to functionally validate regions of interest. T.A.M. performed ChIP experiments. S.W. and D.R. performed RNA-seq. S.W., D.R., L.M.C. and L.J. performed analysis of RNA-seq results. M.J.M. performed statistical analysis. P.B. and J.C. performed all mouse work. C.E., D.H. and J.C. performed and analysed immunohistochemical. S.J.C. supervised the work. P.B., L.J., M.J.M., L.M.C. and S.J.C. drafted the manuscript. All authors reviewed and revised the manuscript.

## Additional information

**Competing financial interests:** The authors declare no competing financial interests.

