## [Peer review file · Nature Communications]

Reviewers' Comments:

Reviewer #1 (Remarks to the Author)

This manuscript analyses the potential functional variants at a locus associated with renal cell carcinoma. By using a broad spectrum of bioinformatic, in vitro and in vivo methods the conclude that rs7132434 is the likely functional variant. the paper is well-written, without unnecessary text. The abstract is particularly clear.

This work is highly relevant since it serves to validates a locus associated with a frequent cancer.

The authors elegantly combine in silico with in vitro and in vivo analyses using state of the art methods, leaving little doubt in their conclusion. It would be nice to see some fotografic images of the tumors.

Reviewer #2 (Remarks to the Author)

The manuscript describes identification of SNP within an AP-1 cis-acting element located in the enhancer of a circadian gene BHLHE41 on chromosome 12p12.1, a renal cancer susceptibility locus. The results are novel, important and of interest, the genetic analysis is convincing, however there are several problems with molecular validation of the genetic findings.

1. Location of the investigated 5 SNPs should be marked in Fig. 1.(minor issue)
2. Supplementary Fig.2 is much more convincing in terms of identification of the different binding to the two alleles than main Fig. 2. Data from sup Fig 2a and 2c should be moved to the main figure.
3. All three antibodies: JunB, c-Jun and JunD supershift as shown in Fig 2e and S2b, thus the decision to focus on c-Jun is not justified, and all three transcription factors should be checked by CHIP, as was done for c-Jun in Fig. 2f and s4.
4. The most important issue is that the effects of BHLHE41 on tumor growth in xenografts and expression of downstream genes were tested only using significant overexpression of the protein (20 and 50 fold) as shown in sFig 10d. It is not clear why stable knockdowns (2-3 different shRNAs) were not performed. This is an essential experiment to validate the proposed role of BHLHE41. It is also not shown if the identified SNP correlates with expression of BHLHE41 in any of the RCC cell lines, so that such cell line(s) can be used as experimental model system for the validation of genetic findings.
5. SFig 8 does not show effects of siRNAs, only overexpression. If siRNA were tested, was the level of BHLHE41 decreased sufficiently to grant the conclusion? Besides, growth of cell lines in culture is not always representative of tumor formation in xenografts.
6. In some figures (eg. sFig.8, the numbers of performed experiments are not given. In general the legends should provide more detailed information about how experiments were done. For example, it is not clear from Fig.4 what was done - I am assuming that BHLHE41 was overexpressed and the levels of other mRNAs were measured. But that should be in the legend.

Reviewer #1 (Remarks to the Author): Expert in genome-wide association studies

This manuscript analyses the potential functional variants at a locus associated with renal cell carcinoma. By using a broad spectrum of bioinformatic, in vitro and in vivo methods the conclude that rs7132434 is the likely functional variant. the paper is well-written, without unnecessary text. The abstract is particularly clear.

This work is highly relevant since it serves to validates a locus associated with a frequent cancer.

The authors elegantly combine in silico with in vitro and in vivo analyses using state of the art methods, leaving little doubt in their conclusion. It would be nice to see some fotografic images of the tumors.

We thank Reviewer #1 for their time and thorough assessment of our manuscript. We spent considerable effort crafting a clear and concise narrative in the paper and are glad Reviewer #1 found the manuscript particularly clear. We agree with Reviewer #1 that our functional characterization of 12p12.1 is highly relevant. To provide further information on our xenograft mouse tumors, we have added photos of the tumors from 4 mice injected with ACHN-BHLHE41 and 5 mice injected with ACHN-vector cells as Supplemental Figures.

Reviewer #2 (Remarks to the Author): Expert in renal cancer and functional studies

The manuscript describes identification of SNP within an AP-1 cis-acting element located in the enhancer of a circadian gene BHLHE41 on chromosome 12p12.1, a renal cancer susceptibility locus. The results are novel, important and of interest, the genetic analysis is convincing, however there are several problems with molecular validation of the genetic findings.

We thank Reviewer #2 for their though review of our manuscript and for considering our results to be novel, convincing, and of interest. To address the concerns of Reviewer #2, please see individual responses to the comments below.

1. Location of the investigated 5 SNPs should be marked in Fig. 1.(minor issue)

We thank Reviewer #2 for suggesting we improve Figure 1 by including additional annotation of relevant SNP RS numbers. These changes have been made. In addition, we have changed 'region' to 'area' to avoid confusion, and we have modified the text and figure legend to indicate that area A contains rs7132434 and rs10842707, area B contains rs718314 and rs17383134; and area C contains rs12814794.

2. Supplementary Fig.2 is much more convincing in terms of identification of the different binding to the two alleles than main Fig. 2. Data from sup Fig 2a and 2c should be moved to the main figure.

Thanks for the suggestion. We have accordingly swapped Supplementary Figure 2 with Figure 2 in the main text.

3. All three antibodies: JunB, c-Jun and JunD supershift as shown in Fig 2e and S2b, thus the decision to focus on c-Jun is not justified, and all three transcription factors should be checked by CHIP, as was done for c-Jun in Fig. 2f and s4.

We appreciate the reviewers point about binding of components of AP-1, c-Jun, JunD and possibly JunB. Given the apparent differences in binding, rather than focusing on c-Jun, we have modified the text to focus on AP-1 as the transcription factor that binds to rs7132434. We chose to perform CHIP only with anti-cJun, because the TCGA data indicates that JunD expression is significantly lower in kidney tumors, than that of c-Jun, and the super shift we observed with anti-JunB was very weak. For these reasons we felt the c-Jun antibody offered the greatest potential to detect AP-1 binding. In addition, the c-Jun antibodies we used for CHIP, were raised against less well conserved regions of c-Jun, and therefore likely cross-react with JunD.

4. The most important issue is that the effects of BHLHE41 on tumor growth in xenografts and expression of downstream genes were tested only using significant overexpression of the protein (20 and 50 fold) as shown in sFig 10d. It is not clear why stable knockdowns (2-3 different shRNAs) were not performed. This is an essential experiment to validate the proposed role of BHLHE41. It is also not shown if the identified SNP correlates with expression of BHLHE41 in any of the RCC cell lines, so that such cell line(s) can be used as experimental model system for the validation of genetic findings.

We chose overexpression of *BHLHE41* for the xenograft model, because TCGA data shows that *BHLHE41* expression is increased in RCC. While, we agree with Reviewer #2 that stable knockdowns would be a good experiment to further characterize the effect of *BHLHE41* in renal cells, construction of stable *BHLHE41* knockdowns and further xenografts are practically beyond the scope of the current paper. (Please see response to comment 5 below for additional information about siRNA knockdown of *BHLHE41*.)

We did not observe an eQTL in renal cell lines. We only have 2 normal renal cell lines (HK2 and RPTEC-hTERT) and both are heterozygous for rs7132434. In the renal cancer cell lines, there are many other mutations, some known, p53, VHL, Hif1a, Hif2a, and many unknown; and they have been selected for growth in culture – where we don't see an effect for *BHLHE41*. Because altering *BHLHE41* expression in cells lines did not produce an observable phenotype, we performed xenografts with *BHLHE41*-overexpressing cells.

5. SFig 8 does not show effects of siRNAs, only overexpression. If siRNA were tested, was the level of BHLHE41 decreased sufficiently to grant the conclusion? Besides, growth of cell lines in culture is not always representative of tumor formation in xenografts.

We did perform *BHLHE41* siRNA knockdown and assessed mRNA and protein levels by qPCR and western blot. We originally did not include this data in the figures, but at the request of Reviewer #2 we are now adding this to the Supplementary Figures. Furthermore, we agree with Reviewer #2 that cell lines in culture are not representative of tumor formation.

6. In some figures (eg. sFig.8, the numbers of performed experiments are not given. In general the legends should provide more detailed information about how experiments were done. For example, it is not clear from Fig.4 what was done - I am assuming that *BHLHE41* was overexpressed and the levels of other mRNAs were measured. But that should be in the legend.

We appreciate the suggestion to make the figure legends more informative. We have added additional information to figure legends including number of experiments performed and details on how experiments were done. In particular, we have clarified the legend for Figure 4 (now Fig. 5) to indicate that *BHLHE41* was overexpressed and the levels of other mRNAs were measured.